# Prevalence of scabies and associated factors among school age children in Qoloji IDP in Babile district, Somali, Eastern Ethiopia

**Ahmed Mohammed Ibrahim**[1]*, **Abebe Belay Reta**[1], **Mohyadin Abdullahi Ahmed**[1], **Ramadan Budul Yusuf**[1] **Abdilahi Ibrahim Muse**[1], **Mohamed Omar Osman**[1], **Seid Muhumed Abdilaahi**[2], **Mustafe Abdi Ali**[3], **Kadar Ahmed Bile**[1]

**1** Department of Public Health, Institute of Health Science, Jigjiga University, Jigjiga, Ethiopia, **2** Department of Pediatrics and Child Health Nursing, Institute of Health Science, Jigjiga University, Jigjiga, Ethiopia, **3** Department of Statistics, College Natural and Computational Science, Jigjiga University, Jigjiga, Ethiopia

\* ahmey114baba@gmail.com, ahmedmohammed@jju.edu.et

## Abstract

### Background

Scabies is a contagious ectoparasite of the skin. It is caused by the mite Sarcoptes scabiei var. Scabies occurs worldwide among people of all ages, races, genders and social classes and has been identified as a neglected tropical infectious disease. In Ethiopia, there is currently social unrest, conflict, and human-made disasters, leading to the displacement of the population from one place to another. Scabies is one of the most common skin diseases among the internally displaced population, where hygiene and sanitation are poor. The aim of the study was to assess prevalence of scabies and associated factors among school age children in Qoloji IDP in Babile District, Somali, Eastern Ethiopia.

### Methods

A cross-sectional community-based study design was carried out in 422 among school aged children from June to July 2022. The data was collected by using observation (clinical investigation), structured questionnaires. Data was coded, entered and cleaned using with epi data version 3.1, and exported and analysis was done by using SPSS statistical software package version 22. Logistic regression analysis was used to identify factors associated with scabies. Findings were presented by using 95% CI of Crude Odds Ratios (COR) and Adjusted Odds Ratios (AOR). To declare statistical significance, a p-value of less than 0.05 was used.

### Results

There were 63 cases of scabies with a prevalence of 14.92% CI at 95% (11.7%-18.1%). The multivariable logistic regression shows that age categories with 5-9 [AOR = 2.4 (95% CI: 1.01,6.39)], over-crowding index greater than 1.5 [AOR = 10 (95% CI: (3.6,27.8)] washing

**Data availability statement:** All data are in the manuscript and/or Supporting Information files (S2 minimal dataset).

**Funding:** The author(s) received no specific funding for this work.

**Competing interests:** The authors have declared that no competing interests exist

**Abbreviations:** AOR, adjusted odds ratio; APSGN, acute post streptococcal glomeruli nephritis; COR, crude odds ratio; DTM, displacement tracking matrix; GBD, Global Burden of Disease; HH, household; HC, host community; IOM, International Organization for Migration; IDP, internal displaced peoples, Institutional Review Board; MDA, Mass Drug Administration; NTD, Neglected Tropical Disease; SPSS, Statistical Package Social Science; UNHCR, United Nation High Commission for Refugees; WASH, water sanitation and hygiene; WHO, World Health organization.

clothes infrequently [AOR = 14.7(95% CI: 3.6,25)), history of contact with scabies patients in the past 2 months [AOR = 5(95% CI: (1.2,23.0)], and Family having rash in the past two weeks [AOR = 9.9 (95% CI: 3.7,26)] having poor knowledge about scabies among children's family [AOR = 8.9 (95% CI: 3.3,24)] were significant variables at p-value less than 0.05.

## Conclusion

There was a higher prevalence of scabies in the study area. Age, overcrowding, washing clothes infrequently, history of contact with scabies patients in the past 2 months, family having a rash in the past two weeks, and poor knowledge about scabies among children's families were significant factors. Health education on personal hygiene, avoiding sharing clothes with others, avoiding contact, sleeping with scabies-ill people, and sharing beds with others are important measures in reducing these risk factors.

## Introduction

Scabies is one of the most predominant global public health problems and neglected parasitic diseases [1]. Worldwide, the prevalence of scabies was about 204 million cases, with 0.21% of total disability-adjusted life years lost, and, in resource-poor tropical settings, the sheer burden of scabies infestation and their complications impose a major cost on healthcare systems [2]. It is usually only considered to be a disease of public health importance in kids, the elderly, indigenous populations, and immune-compromised individuals such as those living with HIV/AIDS [3].

It is also a type of chronic ectodermal skin disease characterized by a prodigious eruption produced by mites, which live and reproduce in the superficial layers of the epidermis [4]. Mite is the common causative agents of this communicable infectious disorder of the skin [5]. The diagnosis is defined as "visible mite under dermatoscopy plus clinical symptoms suggestive of scabies [6]. Typically, prolonged direct skin-to-skin contact with someone who had scabies or sharing the clothes, linens, or bedding that have been used by somebody with scabies, the disease can also spread covertly [7]. Moreover, across the world in resource-poor countries, poverty, low socioeconomic status, and the presence of head lice were identified as determinant factors of scabies [8]. Scabies symptoms impact practically every region of the body; however, the interdigital spaces, wrist, elbow, armpit, penis, nipple, waist, buttock, abdomen, shoulder locations, feet, and thigh are the most commonly affected areas [3]. The scabies parasite is the most common among the disadvantageous population, displaced people, and refugees [9].

Equally both sexes are affected; the epidemiology of scabies in ethnic differences is most likely to be related to differences in socioeconomic, overcrowding, housing, and behavioral factors rather than racial origin [10]. According to the World Health Organization, there were approximately 26.4 million internally displaced people (IDPs) worldwide in 2011. (IDPs) are not afforded the same international protections as refugees under UN law, yet they have been proved to be just as vulnerable [4].

There are several key interventions that can help control the spread of scabies infection, including providing enough safe water for personal hygiene, such as washing clothes, washing the body with soaps, especially the affected areas, and treating individuals with 5% ivermectin with close clinical active case findings [11].

According to many studies, mass ivermectin administration in the community is an important method for the control of scabies, and it should be followed by aggressive case discovery for the reduction in the prevalence of both scabies and bacterial skin diseases [12–14].

Globally, the prevalence of scabies reaches a range of 0.2% to 71.4% in the world [8]. Scabies directly affects over 200 million people, leading to about 21 in 1000 disability-adjusted life years. It is the most common skin problem and affects about 200 million people each year worldwide [6]. Globally, scabies is a common cause of itching dermatosis, infesting around 300 million people [8]. Scabies is most prevalent in tropical low- and middle-income countries, such as East Asia, Southeast Asia, Oceania, and tropical Latin America. In poor countries, the prevalence of scabies ranges from 0.4 to 31% of the population [15]. In many nations, scabies is a substantial public health concern that results in significant morbidity due to secondary bacterial skin infections that cause cellulitis, impetigo, and other dangerous systemic diseases consequences [16].

Ethiopia's government developed integrated and multi-level interventions for scabies prevention and reduction, including advocacy, social mobilization, and social and behavioral change communication at many levels [11]. As a result, determining the prevalence of scabies and its factors among schoolchildren is important to the control of skin diseases and the prevention of scabies outbreaks by updating the disease's burden and prevalence in different study designs and in different seasons in various Ethiopian geographic locations.

There were limited community-based studies on the prevalence of scabies and related factors, despite the fact that there were few studies on the frequency of scabies among schoolchildren elsewhere in Ethiopia. Additionally, nothing was known about the epidemiology and prevalence of scabies in the study area. In order to provide a solution to this issue, the main goals of this study were to identify the incidence of scabies and its related factors among school-aged children in the internally displaced population of Qoloji Camp, Babile District, eastern Ethiopia, and to offer a strategy to address this problem.

Understanding the prevalence of scabies aids in the development of community-based health education that includes anti-scabies messages, construct successful diagnostic and community-based preventative strategies. There are fewer studies focused on scabies in the IDP regarding scabies prevalence and associated risk factors among schoolchildren in our study area. Therefore, this study aimed to assess the prevalence of scabies and associated factors among school-age children in Qoloji IDP in Babile District, Somalia, Eastern Ethiopia.

## Method and materials

### Study setting and period

The study was conducted in Qoloji IDP camp, located within Babile district, which is part of Fafen Zone in the Somali Regional State of Ethiopia. Qoloji is the largest of Ethiopia's numerous settlements for internally displaced persons (IDPs) and is situated about 50 km northeast of Jigjiga city, 60 km from Harar, and 557 km from Addis Ababa. Established in 2016, Qoloji is located in Babile Woreda, near the border between the Oromia and Somali Regions, along the highway between Harar and Jigjiga.

In the Somali Region, most displaced people reside in spontaneous sites or camps, while others live within host communities. Qoloji is the largest displacement settlement in the region, with a population of nearly 80,000 as of 2020 (IOM 2020). The primary IDP groups in Qoloji are Somali communities previously residing in the Oromia Region, displaced due to conflict. Between 2016 and 2020, the settlement expanded by approximately 0.19 km southward each year, currently hosting around 12,834 households, or 79,148 individuals, as per Qoloji Spatial Profile (June 2021). The study was conducted from April 1, 2022, to July 1, 2022.

## Study design

A community based cross-sectional study design was employed.

## Study population

Sample of all school age children aged 5-14 years in the selected kebele in Qoloji IDP who fulfill selection criteria during the study period.

## Eligibility criteria

**Inclusion criteria.** All school age children living in the IDP selected households during study period were included and the one who give consent to participate.

**Exclusion criteria.** The care givers who have difficulty of hearing or speaking and mothers who are severely sick for interview were excluded from the study.

## Sample size determination

The sample size was determined by using single population proportion formula with the assumption of 95% confidence interval (CI), marginal error (d) of 5%, by taking prevalence of scabies 50%, (since there was no study done in Ethiopia among school age children in IDP), none response rate of 10%.

Sample size determination for First objectives.

The sample size was calculated by using single population proportion determination formula.

N = total number of populations=24,694.

The data obtained from Ethiopia neglected tropical diseases (NTD) Master plan 2021, School age children (5-14 years) was 31.2% from the total population that means 31.2% of 79,148 is school age children at the study area [17].

$$n = \frac{Z_{-/2}pq}{d^2}$$

Where;

n = minimum sample size for statistically significant survey.

Z = normal deviant at the portion of 95% confidence interval two tailed test = 1.96.

P = considering the prevalence of scabies (50%).

q = 1−p

d = margin of error acceptable is taken as 5% =0.05.

$$\text{Sample size}(n) = \frac{(1.96)^2 \ (0.5) \ (0.5)}{(0.05)^2} = 384.$$

After adding 10% of non-response rate the final size was 422.

## Sampling procedure

A simple random sampling method was used to select the kebele of Qoloji IDP. The households with school-age children aged 5–14 were selected by using systematic random sampling technique. There were 37 kebeles in Qoloji IDP, and from those 37 kebeles, 10 kebeles were selected randomly, and the number of children aged 5–14 years in the selected kebeles was obtained from the total population. The sample size was then distributed proportionately, taking into account the population size, to each of the chosen kebeles. Subsequently, the K value

was calculated for every seven households using the systematic random sampling approach, which was based on household numbers. The first household was selected by lottery. All school-age children in randomly selected kebele were included in the study. Because there was no data to know which households had eligible children aged 5–14 years for the study, a survey was conducted before two weeks of the data collection to identify eligible children aged 5–14 years households by using household lit formats. When the home contained more than one authorized respondent, only one responded by the lottery method. Two consecutive revisits were arranged when the eligible children were not present during the data collection period. (Fig 1).

## Data collection method and material

The data was collected using a closed-ended structured questionnaire prepared in English and translated into Af-Somali. Trained health professionals conducted the data collection by interviewing parents of children in their homes. Interviewers performed physical examinations on individuals with vesicular skin rashes, itchy, along with clinical observations. To ensure consistency and completeness, daily checks of the questionnaires were conducted.

## Data quality control

To ensure quality of data, questionnaires were prepared in English, translated to Af-Somali, and retranslated back to English by another person who can speak both languages for its consistency. To make sure that the questionnaire was appropriate and understandable, a pre-test was done on 5% of the sample size in the closest kebeles other than the selected ones for the study. The goals, protocols, and methods for gathering data for the project were covered in a one-day training session for the supervisor and data collectors. A multi-sectoral scabies outbreak emergency response document from the Federal Ministry of Health was used to conduct a clinical assessment of all school-age children from a selected household. Monitoring is done on a regular basis while the data is being collected.

## Operational definition

Scabies: in this study scabies is defined as the presence of persistent pruritic rash with itching increasing at night which are notified at least at two specific body sites (on the wrist, sides and web spaces of the fingers, the axillae, peri areolar, per umbilical, genitalia area, abdomen, and buttock areas) with or without history of pruritus in the close entourage [11].

IDP: Displaced people are forced to relocate to a different region of the country due to war and social unrest.

School-age children: children who were in the age group 5-14 years old [11].

Good knowledge: Those mother/care giver who answered above the mean of the knowledge questions [11].

Poor knowledge: Those mother/care giver who answered below the mean of the knowledge questions [11].

Infrequent bathing: showering frequency less than once per week in the past one month [18].

Infrequent washing clothes: washing clothes less than once per week in the past one month [18].

Infrequent changing clothes: changing clothes less than once per week in the past one month [18].

Overcrowding index: is calculated by dividing the number of usual residents in a house by the number of bedrooms in the house. If it was more than 1.5 overcrowded and if it was less than or equal to 1.5 not overcrowded [19].

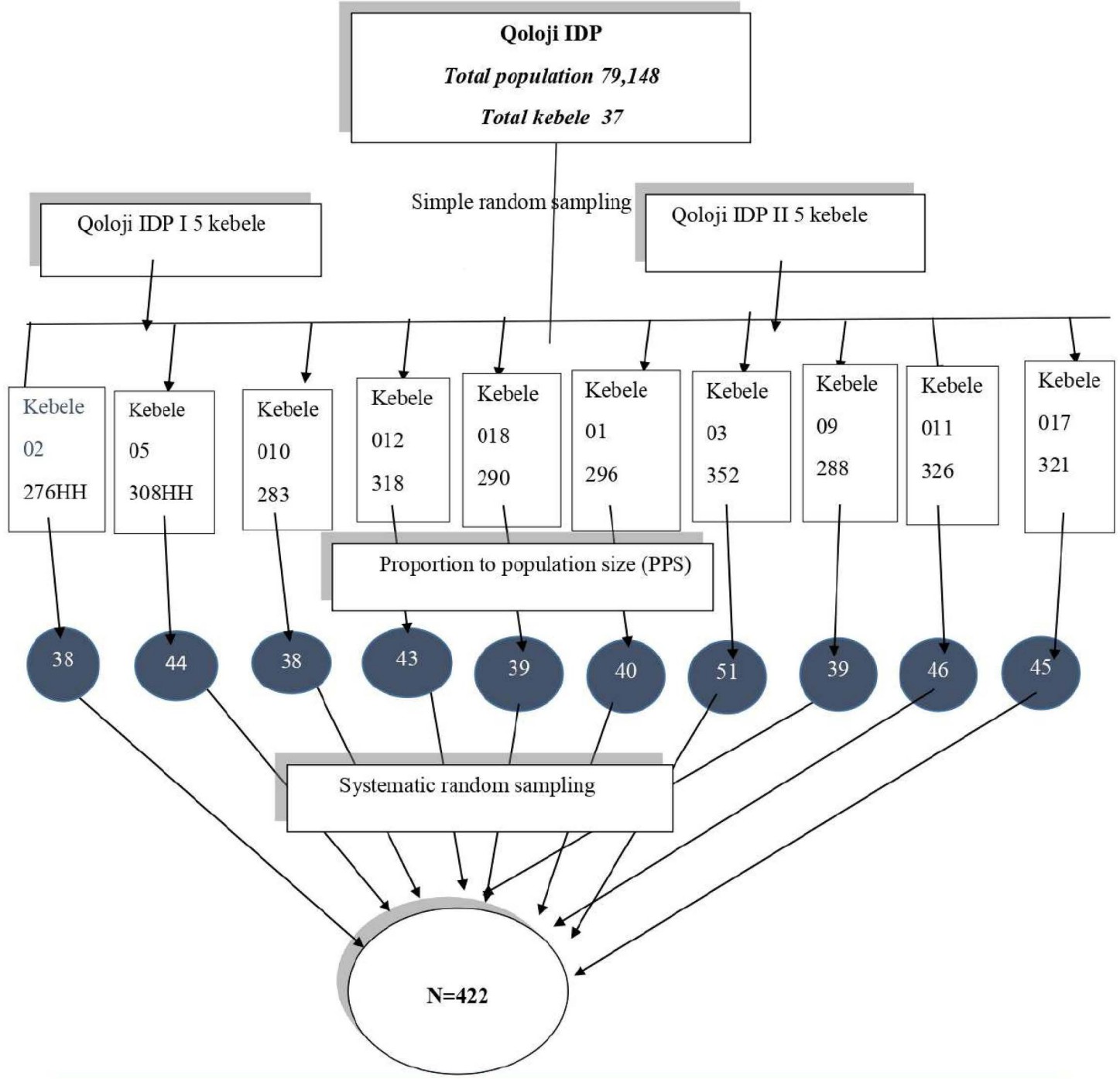

**Fig 1. Schematic diagram of sampling frame among school age children of Qoloji IDP, Somali, Ethiopia, 2022.**

## Data processing and analysis

The data was coded, entered, and cleaned using Epi Data version 3.1 and exported to SPSS statistical software version 22 for analysis. In the univariate analysis, a descriptive statistic was employed to explore the frequency distribution, central tendency, and overall distribution of

independent variables. Bivariate analysis was done to determine the associations between each independent variable and outcome variable. All associated factors with p-value less than 0.25 during bivariate analysis and factors were entered into the multivariable logistic regression model. Odds ratio with 95% confidence intervals was used to see the strength of association between different variables. P-value and 95% confidence interval (CI) for odds ratio (OR) are used in deciding the significance of the associations All covariates that were significant at p value < 0.05 in bivariable analysis were considered for further multivariable analysis to control for all possible confounders and to identify predictors of scabies. The level of statistical significance was declared at a p-value less than 0.05. Finally, variables found to be significant at a p value of 0.05 in the final model were declared as predictor variables. Crude odds ratios (COR) and adjusted odds ratios (AOR) with a 95% confidence interval were reported in the result.

### Ethical considerations

The Jigjiga University Institution of Research and Ethics Review Committee has granted ethical approval for this study (JJURERC0145/2022). The Qoloji IDP administration received the formal authorization letters that the College of Health Science had produced. Each study participant's legally appointed representative provided written informed consent. In order to maintain study participant confidentiality, no identifying information was included in the questionnaires, and participants were made aware that the data would only be utilized for research.

## Results

### Socio-demographic characteristics

A total of 422 participants were included in the study, making a response rate of 100%. More than half (217 (51.5%) of participants were female. The proportion of children aged 5-9 was 48 (76%), and the proportion of children aged 10-14 was 15 (24%). The mean age of study participants was 8.5 years with a standard deviation of ± 2.7 years. More than half of the respondents (219 (51.9%) of the study subjects attended school, whereas 203 (48.1%) did not. The majority (354 (83.9%) of children's mothers interviewed in this study were housewives. Regarding the child's father's occupation, 57 (13.5%) of the children's father's daily laborers, 145 (34.4%) merchants, 30 (7.1%) governmental employed, and 290 (45%) had no jobs recently (Table 1).

### Home environment characteristics

More than three-quarters of the sample, 336 (79.6%), had a family size greater than or equal to five, and the mean family size of households in the district was 5.7 with SD ± 1.3. The majority of 396 (93.8%) of the respondents use wells or springs as a source of water for personal hygiene in the study area. In case of contact history, 31 (7.2%) of the study participants have a history of contact with previous scabies patients in the household. Almost two-thirds, 329 (78%) of study participants, did not gain adequate water supply for personal hygiene and cooking (Table 2).

### Personal hygiene related characteristics of study participants

In terms of personal hygiene, 315 (74.6%) of the participants reported frequent body washing, while 107 (25.4%) reported seldom body washing. Of the participants, 298 (70.6%) wash their clothes frequently, while 124 (29.4%) wash them occasionally. While 136 (32.2%) of the respondents changed their clothes occasionally, more than two-thirds, or 286

**Table 1. Socio-demographic characteristics of study participants in Qoloji IDP, Babile district Eastern Ethiopia, 2022, n = 422.**

| Variables | | Categories | Frequency | Percentage |
|---|---|---|---|---|
| Sex of the children | | Male | 205 | 48.6 |
| | | Female | 217 | 51.4 |
| Age group | | 5–9 years | 247 | 58.5 |
| | | 10–14 years | 175 | 41.5 |
| Children attending school | | Yes | 219 | 51.9 |
| | | No | 203 | 48.1 |
| Family educations | Childs mother | No formal education | 285 | 67.5 |
| | | Primary education | 137 | 32.5 |
| | Childs father | No formal education | 128 | 30.6 |
| | | Primary education | 265 | 62.8 |
| | | Secondary education and above | 27 | 6.4 |
| Occupational status of family | Childs mother | House wife | 354 | 83.9 |
| | | Merchant | 60 | 14.2 |
| | | employed gov't/NGO | 8 | 1.9 |
| | Childs father | Daily laborer | 57 | 13.5 |
| | | Merchant | 145 | 34.4 |
| | | employed gov't/NGO | 30 | 7.1 |
| | | Recently no jobs | 290 | 45 |

**Table 2. Home environment characteristics of study participants in Qoloji IDP, Babile district, Eastern Ethiopia, 2022; N = 422.**

| Variables | Categories | Frequency | Percentage |
|---|---|---|---|
| Family size | < 5 | 86 | 20.4 |
| | >=5 | 336 | 79.6 |
| Sharing bed | Yes | 419 | 99.3 |
| | No | 3 | 0.7 |
| Contact history | Yes | 31 | 7.2 |
| | No | 391 | 92.7 |
| Animals at home | Yes | 120 | 28.4 |
| | No | 302 | 71.6 |
| Have you infected by scabies before? | Yes | 68 | 16.1 |
| | No | 354 | 83.9 |
| Source of water for personal hygiene | Pipe/tap water | 16 | 3.8 |
| | River/pond | 6 | 1.4 |
| | Well/spring | 396 | 93.8 |
| Knowledge status | Good | 246 | 58.3 |
| | Poor | 176 | 41.7 |
| Distance of water source from home | Near home/<=30 minute | 362 | 85.7 |
| | Far away from home > 30 minute | 60 | 14.3 |
| Adequate water supply | Yes | 93 | 22 |
| | No | 329 | 78 |

(67.8%), of the respondents changed into clean clothes frequently. In terms of how frequently the youngsters wash their hair, 310 (73.5%), 71 (16.8%), and 9.7% (41) wash it every 1-3 days, 7-14 days, and more than 14 days, respectively. While 107 (25.42%) of participants did share their clothes with anyone else. Of the children, 112 (26.5%) had their fingernails trimmed or short, while about 310 (73.5%) did not. When it comes to showering, almost two-thirds of students 360 (85.3%) use soap, whereas only 62 (14.7%) pupils utilize water alone. A skin rash history ran in the family for about 64 (15.2%) of study participants, while 358 (84.8%) did not (Table 3).

## Knowledge about scabies among children's family

Of all the respondents, 285 (67.8%) were aware of the symptoms and signs of scabies, 280 (66.4%) were aware of the etiology of scabies, and 73 (40.8%) were aware of the body parts affected, including the finger webs, armpits, genitalia, abdomen, breast, waist, and knees. Fifty-seven (30.7%) were aware that scabies primarily affects covered body parts, and twenty-five (25.6%) stated the genitalia area. (Fig 2).

Regarding scabies prevention, nearly half 139 (47%) indicated frequent bathing and avoiding physical contact with scabies patients, followed by 88 (29.7%) mentioned frequent bathing and cleaning of clothes simply. Lastly, over three-quarters 304 (72%) of the participants claimed that scabies can be avoided by drying pillows and mattresses (Table 4).

## Prevalence of scabies

The overall prevalence of scabies in Qoloji IDP Babile district was 14.92% CI 95% of 11.7%-18.1%. From the total 63 scabies cases, 76.2% were found between the age groups 5-9 years, and 23.8% were found between the age groups 10-14 years (Fig 3).

**Table 3. Personal hygiene and sanitation characteristics of study participants in Qoloji IDP, Babile district Eastern Ethiopia, 2022; N = 422.**

| Variables Category | | Frequency | Percentage |
|---|---|---|---|
| Body washing | Frequently | 286 | 67.8 |
| | Infrequently | 136 | 32.2 |
| Washing clothes | Frequently | 274 | 64.9 |
| | Infrequently | 148 | 35.1 |
| Changing of clothes | Frequently | 286 | 67.8 |
| | Infrequently | 136 | 32.2 |
| Frequency of hair washing | 1–7 days | 300 | 71.1 |
| | 7–14 days | 72 | 17.1 |
| | >14 days | 50 | 11.8 |
| Share clothes with other | Yes | 107 | 25.4 |
| | No | 315 | 74.6 |
| History of skin rash | Yes | 67 | 15.9 |
| | No | 355 | 84.1 |
| Finger nails | Clean | 367 | 87 |
| | Dirty | 55 | 13 |
| Fingernails cut short/trimmed | Yes | 310 | 73.5 |
| | No | 112 | 26.5 |
| Use soap for washing and showering | Yes | 360 | 85.3 |
| | No | 62 | 14.7 |

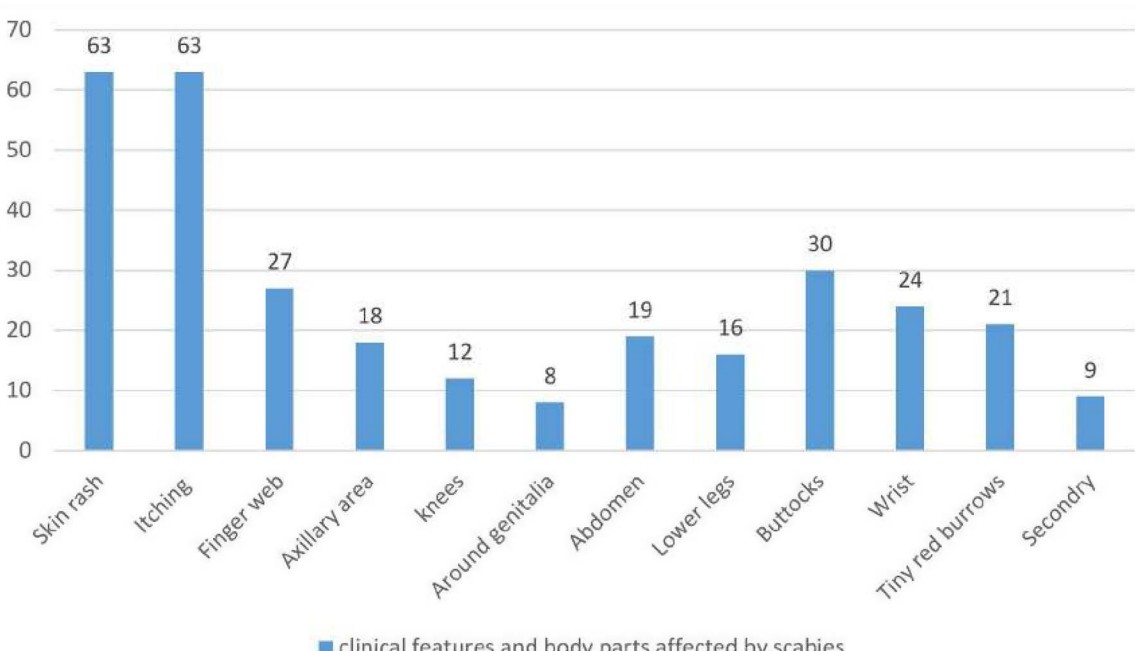

**Fig 2. Cases with clinical features of scabies, Qoloji IDP, Babile district, eastern Ethiopia, 2022.**

## Factors associated with scabies disease

All variables that had a p-value of 0.25 in bi-variable analysis were inserted in multivariable logistic regression and included in the logistic regression model. After adjusting for these variables, age of children with age categories, overcrowding index, history of contact with scabies patients in the past 2 months, infrequently washing clothes, sharing clothes with others, and knowledge of about scabies were significantly associated with scabies disease. Children with age group 5-9 were 2 [AOR = 2.4, 95% CI: 1.01, 6.39] times higher than those who were in age group 10-14. In comparison to overcrowding index less than or equal to 1.5, those with an overcrowding index more than 1.5 had a 10 [AOR = 10, 95% CI: 3.6, 27.8] times increased risk of contracting scabies. When compared to children who had no family history of scabies or who had never had contact with scabies patients, those who had a family member with scabies or a history of contact with scabies patients during the previous two months were four [AOR = 4, 95%CI:1.2,23.0] times more likely to contract scabies. Compared to children who routinely washed their garments, those who did not had a 15 [AOR = 14.7, 95% CI: 3.2, 65.7] times higher risk of scabies illnesses. According to this study, kids who shared clothes with other kids had a 10 [AOR = 9.9, 95%CI: 3.7, 26] times higher chance of getting scabies than kids who didn't share clothes. Youngsters were nine [AOR = 8.9, 95% CI: 3.3, 24] times more likely to contract scabies than children whose families had strong understanding of the disease (Table 5). The Hosmer-Lemeshow test was done, and the p-value was 0.994, which indicates the model is adequately fitted.

## Discussion

In Qoloji IDP, the overall scabies prevalence in this survey was 14.92% at CI 95% (11.7%-18.1%). Information about the frequency of scabies and its contributing factors in school-age children is useful in developing prevention, control, and rehabilitative care strategies.

**Table 4. Knowledge about scabies among children's family characteristics of study participants in Qoloji IDP, Babile district Eastern Ethiopia, 2022 N = 422.**

| Variables | Categories | frequency | Percentage (%) |
|---|---|---|---|
| Have you Ever heard about scabies? | Yes | 391 | 92.7 |
| | No | 31 | 7.3 |
| Do you know Etiology of the scabies | Parasite | 107 | 38.2 |
| | Germs | 76 | 27.1 |
| | The effect of scratching | 97 | 34.6 |
| Signs and symptoms of scabies | Itchy skin rash worsens at night | 285 | 67.8 |
| | Don't know | 136 | 32.2 |
| Parts of the body that are affected by scabies | Finger webs, armpits, genitalia, buttocks, abdomen, breast, knees | 73 | 40.8 |
| | Parts that are mostly covered | 55 | 30.7 |
| | Mostly at genitalia | 51 | 28.5 |
| Transmission way | Skin to skin contact and through contaminated fomites | 140 | 43.9 |
| | Through skin contact only | 113 | 35.4 |
| | Through fomites only | 66 | 20.7 |
| Who is Sufferer from scabies? | All age group but mostly children | 153 | 50 |
| | Teenagers only | 106 | 34.6 |
| | Only in a certain age groups | 47 | 15.4 |
| Exchanging clothes spread scabies? | Yes | 314 | 74.4 |
| | No | 108 | 25.6 |
| Ways to break the chain of scabies transmission | Disinfect fomites and give treatment | 121 | 38.4 |
| | Keep distant from scabies patient | 128 | 40.6 |
| | Need regular treatment only | 66 | 21 |
| Drying mattress and pillows prevent scabies | Yes | 304 | 72 |
| | No | 118 | 28 |
| Prevention measure for scabies. | Frequent bathing and avoid physical contact with scabies patient | 139 | 47 |
| | Frequent bathing and cleanliness of clothes only | 88 | 29.7 |
| | Keep fomites from contamination only | 69 | 23.3 |
| Knowledge status of children's family | Good | 176 | 41.7 |
| | Poor | 246 | 58.3 |

Comparable with study conducted in Cameroon 17.8% [19], Nigeria 13.3% [20], Arbaminch district, south part of Ethiopia 16.4% [11], Raya Alamata, Ethiopia which is 12.93% prevalence [5].

However, this study was lower than study done at Pakistan with prevalence of 47.6% [21], Malaysia, 31% [22], Sierra Leon displacement camp, 25% [23], Solomon Islands, 19.2% [24], Fiji, 18.5% [16], and Wadila district, in northern Ethiopia, 23.8% [8]. This variation may result from variations in the research duration, sample size, knowledge and health-seeking behavior between these groups, and the type of evaluation (microscopic or clinical).

But prevalence of this study is higher than the research done in Egypt with a prevalence of 4.4% [25], Dabat district in northern Ethiopia, 9.3% [26]. These differences may result from differences in knowledge, water accessibility, hygiene and sanitation practice and time of study differences, since scabies is sometimes seasonal and get higher in winter season, and gets

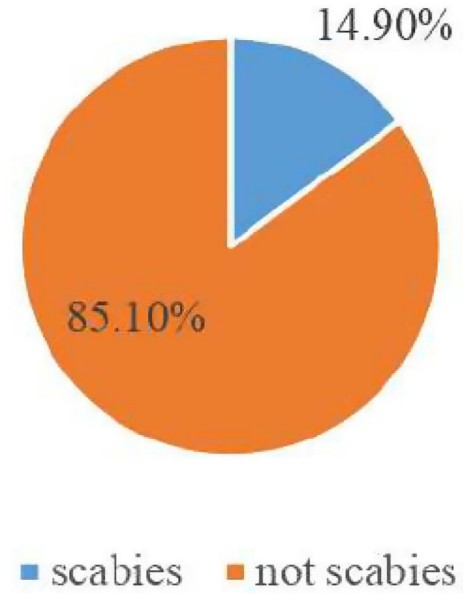

**Fig 3. Scabies among school age children in Qoloji IDP, Babile district, 2022, N = 422.**

low during rainy or summer season and also as the study time is earlier study participants may get sensitization about hygiene and sanitation.

Children in the age group of 5–9 years old were more affected by scabies than those children aged 10–14 years old. Study participants who were in age group 5-9 years old were 2.4 times more likely to develop scabies diseases compared to children in age group 10-14 years old, all other predictors being fixed at AOR = 2.4; 95% CI [1.01, 6.39]. This study is in line with the study in Australia; children in the age group 5–9 were 3.7 times more likely than other age groups to develop scabies diseases (AOR = 3.7, 95% CI [3.0, 4.4]) [16]. It is also supported by another study done in Malaysia that showed that the odds of scabies infection with the age of 5-9 years old were 2 higher than other age groups (AOR = 2.3, 95% CI [1.27, 7.69]) [22].

The study done in Fiji revealed that children 5–9 years old were 2 times more highly infected by scabies diseases than other age groups. AOR = 2.2; 95% CI [1.9, 2.7]. This is due to the fact that children have no strong immunity; as a result, they are easily affected by communicable diseases like scabies, and they cannot protect their hygiene and sanitation. Most of the time, this age group is exposed to malnutrition, and children are susceptible to infectious diseases since scabies is a disease of disadvantaged populations. But Another study done in Ethiopia from Raya Alamata district in 2018 on an age group of 10–14 children was 2.3 times more likely to be infected by scabies (AOR = 2.3, 95% CI [0.23, 6.20]) [5]. This might be due to the difference in community-based and facility-based study design because most children start schooling above ten years in our rural setup, so during study there may be more student's age greater than 10 years old.

According to various studies, one of the primary risk factors for scabies infection in Ethiopia and other regions of the world is overcrowding. Additionally, overcrowding index was found to be positively associated with scabies in this analysis; overcrowding index greater than

**Table 5. Multivariable logistic regression analysis of factors associated with scabies among school age children in Qoloji IDP, 2022.**

| Variables | Categories | Scabies | | COR [95%CI] | AOR [95%CI] | P value at 0.05 |
|---|---|---|---|---|---|---|
| | | Yes | No | | | |
| Age | 5–9 | 48 | 199 | 2.5 [1.4–4.7] | 2.4 [1.01–6.39] | 0.048 |
| | 10–14 | 15 | 160 | 1 [Ref] | [Ref] | |
| Overcrowding index | <=1.5 | 7 | 247 | 1 [Ref] | [Ref] | |
| | >1.5 | 56 | 112 | 17.6 [7.7–39.9] | 10 [3.6–27.8] | 0.000 |
| History of contact with scabies patients past 2 months | Yes | 15 | 16 | 6.6 [3,4,4–13–] | 4 [1.2–23.0] | 0.028 |
| | No | 48 | 343 | 1 [Ref] | [Ref] | |
| Have you infected by scabies before | Yes | 28 | 80 | 2.790 [1.6–4.8] | 2.2 [0.876–5.712] | 0.09 |
| | No | 35 | 279 | 1 [Ref] | [Ref] | |
| Animals at home | Yes | 30 | 112 | 2 [1.1–3.4] | 1.3 [0.543–3.188] | 0.53 |
| | No | 33 | 247 | 1 [Ref] | [Ref] | |
| How often did you wash your body | Frequently | 15 | 271 | 1 [Ref] | [Ref] | |
| | Infrequently | 48 | 88 | 9.85 [5.2–18.4] | 0.83 [0.196–0.352] | 0.8 |
| How often did you wash your clothes | Frequent | 9 | 265 | 1 [Ref] | [Ref] | |
| | Infrequent | 54 | 94 | 16.9 [8–35.5] | 14.7 [3.2–65.7] | 0.000 |
| How often did you change clean clothes? | Frequent | 32 | 254 | 1 [Ref] | [Ref] | |
| | Infrequent | 31 | 105 | 2.3 [1.3–4] | 1.8 [0.724–4.474] | 0.206 |
| Family having rash in the past two weeks | Yes | 23 | 44 | 4 [2.2–7.5] | 0.5 [0.16–1.85] | 0.334 |
| | No | 40 | 315 | 1 [Ref] | 1 [Ref] | |
| Did you share your clothes with others | Yes | 38 | 69 | 6.3 [3.6–11] | 9.9 [3.7–26] | 0.000 |
| | No | 28 | 290 | 1 [Ref] | [Ref] | |
| How often do you wash your hair | 1–7 days | 39 | 89 | 1 [Ref] | [Ref] | |
| | 7–14 days | 13 | 79 | 1.4 [0.7–2.9] | 1.17 [0.395–3.494] | 0.77 |
| | >14 days | 11 | 202 | 1.8 [0.8–3.9] | 2.47 [0.66–9.18] | 0.17 |
| Knowledge about scabies | Good | 54 | 122 | 1 [Ref] | [Ref] | |
| | Poor | 9 | 237 | 0.086 [0.04,0.18] | 8.9 [3.3–24] | 0.000 |

NB: Significant variable at p-value < 0.25 in bivariate and < 0.05 at multivariate.

1.5 was associated with a 10-fold increased risk of scabies compared to overcrowding index 1.5 and below (AOR = 10, 95% CI [3.6,27.8]). This could include sharing a mattress and blanket within the family and having regular contact with scabies patients.

In terms of personal hygiene, the results of this study showed a favorable correlation between scabies and infrequent clothes washing. Compared to study participants who laundered their garments frequently, those who laundered their clothes infrequently had an almost 15-fold increased risk of developing scabies diseases [3.2, 65.7]. This is in line with research from the Arba Minch District in south Ethiopia, which found a strong correlation between scabies and infrequent clothes cleaning. AOR = 5.4, 95% CI [2.264, 13.04] [11]. This might be due to the fact that the scabies mite may reside in clothes and bedlines for almost a week; this gives the scabies mite a chance to survive and continue spreading from one to another. The other associated predictor was sharing clothes with others. The odds of individuals that share clothes with others were almost 10 times higher among scabies cases compared to those who did not share their clothes with AOR = 9.9; 95% CI [7.3, 26]. This research aligns with those conducted in Wadila, northern Wollo, Ethiopia, and showed that children who share clothes were 10 times more likely to develop scabies. AOR = 10, 95% CI [3.4,30.2] [8].

This study is consistent with the study done in the northern part of Ethiopia, Raya Alamata, which showed that children who share their clothes with others were 18 times more

likely to develop scabies infection (AOR = 17.61, 95% CI [4.98, 62.23]) [5]. Similarly, another study in west Gojjam found that sharing a cloth with a scabies-affected person increased the risk of developing scabies by 3.3 times AOR = 3.3, 95% CI [1.536, 7.149] [27]. This result is also in line with the study done in South Africa and nationalities, East Bedewacho [2]. This might be due to scabies mites that can stay out of human skin for up to two days; physical transmission of the female mites through feces; and also sharing of clothes also shares the parasite to transmit from ill to non-ill persons, and it can increase the rate of infection through time.

This study also revealed that children who had a history of contact with a scabies patient in the past two months were four times more likely to be higher when compared to those without a history of contact (AOR = 4; 95% CI (1.2, 23.0). This outcome was in line with research conducted in the Gamo Gofa Zone, Arbaminch Zuriya Woreda, which revealed that children who had previously interacted with scabies sufferers were 10 times more likely to contract the disease than those who had not. AOR = 9.579 with 95% Confidence Interval [4.03, 17.22] [11]. This might be explained by scabies being one of the communicable diseases that can be transmitted through physical body contact from the infected person to another healthy person.

Children whose family had poor knowledge about scabies AOR = 8.9, 95% CI [3.3, 24] were 9 times more highly affected by scabies disease than family members who had good knowledge. This is consistent with the study in southern Ethiopia, Arbaminch-zuria district, which showed that those with inadequate understanding were 5.2 times more likely than those with enough knowledge to contract scabies. AOR = 5.20, 95% CI [2.188, 12.358] [11].

This study also supported a study done in Damboya, Kembata-Tembaro Zone, revealed that primary school children in Southern Ethiopia's found that individuals with inadequate scabies awareness were 4 times more likely to be infected than those with good knowledge (AOR = 4.32; 95% CI: 2.93, 6.36) [6]. This could be because of the fact that families who had good knowledge about the disease take care for the children and themselves from the disease as well as be treated immediately. This study further revealed that all children diagnosed scabies had itchy skin rash worsen at night. Body parts affected by scabies were finger webs and ulnar area, auxiliary area, genitalia, on the abdomen, on the shoulder blades, on the elbow, on the buttocks and on the lower legs.

## Limitation of the study

The lack of skin scrapings and/or dermoscopy is one of the study's limitations; as a result, false positive results could occur. Therefore, it may be more representative than the study carried out using lab tests. The diagnosis was derived solely from an objective clinical assessment carried out by qualified health officers. Secondly, a cause-and-effect link cannot be established due to the fundamental nature of cross-sectional design. Third, it's possible that not all Ethiopian youngsters of school age will benefit from the findings.

## Conclusion and recommendation

The current study found that prevalence of scabies was high among school-age children in the Qoloji Internally displaced people, which requires public health attention. Age years, residing in overcrowding, family knowledge, infrequent washing of clothes, sharing clothing with others, and a history of contact patients with scabies were associated with scabies. Scabies prevention strategies should be strengthened through health promotion and education about personal hygiene in the community. Treatment for scabies cases and their families and associates should be provided to raise awareness regarding the modes of transmission and control mechanisms. Health education should focus on personal hygiene (frequent bathing, washing

clothes, and regularly changing clothes), refraining the sharing of clothes, limiting contact with scabies-ill individuals, and avoiding sharing beds.

## Supporting information

**S1 File. Data used in analysis for this study.**
(XLSX)

## Acknowledgment

The authors would like to thank everyone who helped with this original paper in any way.

## Author contributions

**Conceptualization:** Ahmed Mohammed Ibrahim, Abebe Belay Reta, Mohyadin Abdullahi Ahmed, Ramadan Budul Yusuf, Abdilahi Ibrahim Muse, Mohamed Omar Osman, Seid Muhumed Abdilaahi, Mustafe Abdi Ali, Kadar Ahmed Bile.

**Data curation:** Ahmed Mohammed Ibrahim, Abebe Belay Reta, Mohyadin Abdullahi Ahmed, Ramadan Budul Yusuf, Abdilahi Ibrahim Muse, Mohamed Omar Osman, Seid Muhumed Abdilaahi, Mustafe Abdi Ali, Kadar Ahmed Bile.

**Formal analysis:** Ahmed Mohammed Ibrahim, Abebe Belay Reta, Mohyadin Abdullahi Ahmed, Ramadan Budul Yusuf, Abdilahi Ibrahim Muse, Mohamed Omar Osman, Seid Muhumed Abdilaahi, Mustafe Abdi Ali, Kadar Ahmed Bile.

**Funding acquisition:** Ahmed Mohammed Ibrahim, Mohyadin Abdullahi Ahmed, Ramadan Budul Yusuf, Abdilahi Ibrahim Muse, Mohamed Omar Osman, Seid Muhumed Abdilaahi, Mustafe Abdi Ali, Kadar Ahmed Bile.

**Investigation:** Ahmed Mohammed Ibrahim, Abebe Belay Reta, Mohyadin Abdullahi Ahmed, Ramadan Budul Yusuf, Abdilahi Ibrahim Muse, Mohamed Omar Osman, Seid Muhumed Abdilaahi, Mustafe Abdi Ali.

**Methodology:** Ahmed Mohammed Ibrahim, Abebe Belay Reta, Mohyadin Abdullahi Ahmed, Ramadan Budul Yusuf, Abdilahi Ibrahim Muse, Mohamed Omar Osman, Seid Muhumed Abdilaahi, Mustafe Abdi Ali.

**Project administration:** Ahmed Mohammed Ibrahim, Abebe Belay Reta, Mohyadin Abdullahi Ahmed, Ramadan Budul Yusuf, Abdilahi Ibrahim Muse, Mohamed Omar Osman, Seid Muhumed Abdilaahi, Mustafe Abdi Ali.

**Resources:** Abebe Belay Reta.

**Software:** Ahmed Mohammed Ibrahim, Abebe Belay Reta, Mohyadin Abdullahi Ahmed, Ramadan Budul Yusuf, Abdilahi Ibrahim Muse, Mohamed Omar Osman, Seid Muhumed Abdilaahi, Mustafe Abdi Ali.

**Supervision:** Ahmed Mohammed Ibrahim, Abebe Belay Reta, Mohyadin Abdullahi Ahmed, Ramadan Budul Yusuf, Abdilahi Ibrahim Muse, Mohamed Omar Osman, Seid Muhumed Abdilaahi, Mustafe Abdi Ali, Kadar Ahmed Bile.

**Validation:** Ahmed Mohammed Ibrahim, Abebe Belay Reta, Mohyadin Abdullahi Ahmed, Ramadan Budul Yusuf, Abdilahi Ibrahim Muse, Mohamed Omar Osman, Seid Muhumed Abdilaahi, Mustafe Abdi Ali, Kadar Ahmed Bile.

**Visualization:** Ahmed Mohammed Ibrahim, Abebe Belay Reta, Mohyadin Abdullahi Ahmed, Ramadan Budul Yusuf, Abdilahi Ibrahim Muse, Mohamed Omar Osman, Seid Muhumed Abdilaahi, Mustafe Abdi Ali.

**Writing – original draft:** Ahmed Mohammed Ibrahim, Abebe Belay Reta, Mohyadin Abdullahi Ahmed, Ramadan Budul Yusuf, Abdilahi Ibrahim Muse, Mohamed Omar Osman, Seid Muhumed Abdilaahi, Mustafe Abdi Ali.

**Writing – review & editing:** Ahmed Mohammed Ibrahim, Abebe Belay Reta, Mohyadin Abdullahi Ahmed, Ramadan Budul Yusuf, Abdilahi Ibrahim Muse, Mohamed Omar Osman, Seid Muhumed Abdilaahi, Mustafe Abdi Ali.

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
