## [Decision Letter · Decision Letter 0]

2 Oct 2024

PONE-D-24-34114PREVALENCE OF SCABIES AND ASSOCIATED FACTORS AMONG SCHOOL AGE CHILDREN IN QOLOJI IDP IN BABILE DISTRICT, SOMALI, EASTERN ETHIOPIA.PLOS ONE

Dear Dr. Ibrahim,

Thank you for submitting your manuscript to PLOS ONE. After careful consideration, we feel that it has merit but does not fully meet PLOS ONE’s publication criteria as it currently stands. Therefore, we invite you to submit a revised version of the manuscript that addresses the points raised during the review process.

We look forward to receiving your revised manuscript.

Kind regards,

Awatif Abid Al-Judaibi, PhD

Academic Editor

PLOS ONE

Journal requirements: 1. When submitting your revision, we need you to address these additional requirements.Please ensure that your manuscript meets PLOS ONE's style requirements, including those for file naming. The PLOS ONE style templates can be found at https://journals.plos.org/plosone/s/file?id=wjVg/PLOSOne_formatting_sample_main_body.pdf and https://journals.plos.org/plosone/s/file?id=ba62/PLOSOne_formatting_sample_title_authors_affiliations.pdf 2. Please include a complete copy of PLOS’ questionnaire on inclusivity in global research in your revised manuscript. Our policy for research in this area aims to improve transparency in the reporting of research performed outside of researchers’ own country or community. The policy applies to researchers who have travelled to a different country to conduct research, research with Indigenous populations or their lands, and research on cultural artefacts. The questionnaire can also be requested at the journal’s discretion for any other submissions, even if these conditions are not met.  Please find more information on the policy and a link to download a blank copy of the questionnaire here: https://journals.plos.org/plosone/s/best-practices-in-research-reporting. Please upload a completed version of your questionnaire as Supporting Information when you resubmit your manuscript. 3. Please provide a complete Data Availability Statement in the submission form, ensuring you include all necessary access information or a reason for why you are unable to make your data freely accessible. If your research concerns only data provided within your submission, please write "All data are in the manuscript and/or supporting information files" as your Data Availability Statement. 4. Your ethics statement should only appear in the Methods section of your manuscript. If your ethics statement is written in any section besides the Methods, please move it to the Methods section and delete it from any other section. Please ensure that your ethics statement is included in your manuscript, as the ethics statement entered into the online submission form will not be published alongside your manuscript. 

Reviewers' comments:

Reviewer's Responses to Questions

**Comments to the Author**

1. Is the manuscript technically sound, and do the data support the conclusions?

Reviewer #1: Yes

Reviewer #2: Yes

Reviewer #3: Yes

2. Has the statistical analysis been performed appropriately and rigorously? 

Reviewer #1: Yes

Reviewer #2: No

Reviewer #3: Yes

3. Have the authors made all data underlying the findings in their manuscript fully available?

Reviewer #1: Yes

Reviewer #2: Yes

Reviewer #3: Yes

4. Is the manuscript presented in an intelligible fashion and written in standard English?

Reviewer #1: Yes

Reviewer #2: No

Reviewer #3: No

5. Review Comments to the Author

Reviewer #1: Journal of PLOS ONE

Title: PREVALENCE OF SCABIES AND ASSOCIATED FACTORS AMONG SCHOOL AGE CHILDREN IN QOLOJI IDP IN BABILE DISTRICT, SOMALI, EASTERN ETHIOPIA.

Manuscript Number: PONE-D-24-34114

Article Type: Research Article

Dear authors,

Thank you for submitting your research article to the Journal of PLOS ONE. I appreciate the effort you have put into conducting this study and sharing this interesting findings with the scientific community. To improve the quality of your research I put some comments and suggestion below.

Reviewer Comments

Abstracts

Comment 1: In Ethiopia now a day there are social unrest, conflict and human made disasters for the sake of this there are displacement of the population from one place to another to relieve this problem and it is the commonest skin diseases of the population lives in internal displaced people. I think it’s well if the authors changes above statements into “In Ethiopia now day there are social unrest, conflict and human made disasters to relieve this problem there are displacement of the population from one place to another”. Also delete the statements “it is the commonest skin diseases of the population lives in internal displaced people and why the authors relate the scabies with conflicts or social unrest, rather its better if the authors relate scabies with hygienic practice, living standards etc. If the research’s conducted in displacement camps, it’s ok.

Comment 1: Dear researchers I could not seen the significance of this study/why this study was needed?. Briefly mention your significance. Also merge your study objective with your background.

Comment 2: The data was collected by using observation (clinical investigation), structured questionnaires.

Comment 3: Data was coded, entered and cleaned using with epi data version 3.1, and exported and analysis was done by using SPSS statistical software package version 22. Rearrange above statements, and shorten “Logistic regression analysis was used to identify factors associated with scabies. Findings were presented by using 95% CI of Crude Odds Ratios (COR) and Adjusted Odds Ratios (AOR). To declare statistical significance, a p-value of less than 0.05 was used”

Comment 4: in conclusion “Health education on personal hygiene (frequent bathing, washing clothes, and changing clothes regularly), avoiding sharing clothes with others, avoiding contact, sleeping with scabies-ill people, and sharing beds with others”. Simply this list of risk factors, why this factors listed here without any messages. I recommended that the authors mentioned why He list them i.e. his recommendation or the practice of community. Keywords must written in alphabet orders.

INTRODUCTION

Dear authors it’s interesting if you add the prevalence of scabies in Ethiopia and the study area and also add information about taxonomy of mites and also mention the group organism in which scabies belongs. Dear authors I have seen in your abstracts the scabies is a disease that do not discriminate age, sex, social class etc., but in your introduction there is contrasting idea make clear why this is possible in the introduction.

In this part many sentence is reported without referenced, add necessary references.

Why the authors write some parts of introduction using different text and fonts.

Comment 5: Line number 50-51 “Mite is the common cause of this communicable infectious disorder of the skin” change into “Mite is the common causative agents of this communicable infectious disorder of the skin

Comment 6: Line number 51-51: And the mites are not noticeable with the naked eye but can be seen with microscope” delete or add to diagnosis information.

Comment 7: Line number 56-57, the onset of the symptoms depends on the level of infection; a scabies patient normally has 12 mites present at any given time. What is 12, is the numbers of mites per infected site/wound or the number of mites on/in the body of infected person? Clarify.

Comment 8: Include Line number 99-111, under the first paragraph and also there is repetition of sentence in your introduction, For example line number 82-85 with line number 112-`13.

Material and Methods

Dear authors what is the difference b/n the target population and study population, I think its good if the authors merge them.

Comment 9: Line number 157: a large majority of displaced persons change the word persons into people and also put the study period at the starts of this parts. Line 171 change the word conducted into employed.

Result

A total of 413 participants were included in the study, making a response rate of 100%. Is your sample size 413 or 422? Make sure. I have seen throughout your result parts the percentage of the response if written before the frequency, also in some area you have written only percentage where as in others both frequency and percentages were reported. Dear authors I recommended you its better if you rewrite the results of your study in similar manners (example 336 (79.6%) or (79.6%), but not possible to write 79.6% (336), although this may not changes what intended to be mentioned, it may reduce the attractiveness and clarity of your paper.

Comment 10: Line 303, more than three-quarters of the sample (79.6%), had a family size greater than or equal to five, rather to say sample, changes to participants, and correct “the majority of 396 (93.8%) of the respondents use wells or springs as a source of water for personal hygiene in the study area” as “the majority of respondents (396 (93.8%) use wells or springs as a source of water for personal hygiene in the study area. Line number 306-307: “Regarding the water source, nearly almost all (93.8%) of households use improved water sources”. Clarify the improved water (i.e. in quality or improved in quantity).

Comment 11: Line number 307-308: 7.2% (31) of the study participants have a history of contact with previous scabies patients in the household. Use only percentages, if you include frequency include in all results, but before the percentages outside of the brackets.

DISCUSSION

I appreciated your discussion and for your time to prepare such well organized and informative discussion. But to make your discussion more interesting I have put some recommendation and correction below.

Comment 12: Line number 316: 14.92 is frequency or percentages/mean and dear authors why you do not use similar significant figures (example 12.93% other hands use 16.4%)

Comment 13: The authors mentioned that your study is comparable with study conducted in Cameroon 17.8% (18), Nigeria 13.3% (19), Arbaminch district, south part of Ethiopia 16.4 (12), Raya Alamata, which is 12.93% (5). Based on what, like your results you have to mention what they reported, here you are list only percentages. Correct as “comparable with study conducted in Cameroon 17.8% (18), Nigeria 13.3% (19), Arbaminch district, south part of Ethiopia 16.4% (12), Raya Alamata, Ethiopia which is 12.93% prevalence (5)

Conclusion and Recommendation

Prepared well.

Generally the document needed some grammar and language improvement

Thank you for considering these suggestions to enhance the quality and clarity of your research article. I look forward to seeing the revised manuscript.

Best regards,

[Abdulaziz Abrahim]

Reviewer for the Journal of PLOS ONE

Reviewer #2: in many place there is typographical errors including the name of Place as example a place that found in the northen parts of Ethiopia called Woldia is not write in its correct name.

1. Some ideas are written without any citation (reference), e.g., line 100; line 125 said according to many studies... However, there is no a mentioned reference... So please insert some selective references and another place, so please revise again all introduction parts.

2. Line 149 rearranges the title, like Materials and Methods.

3. In the title “Study Setting and Period,” you mentioned the study area, so it requires some modification; moreover, you didn’t explain where the Somali region is found in Ethiopia, e.g., distance from the capital of Ethiopia, in which direction, etc.

4. To me, the titles on lines 173 and 175 are the same, so it is better if you merge them. Generally, see again from lines 173-184.

5. Who is the honor of the single population proportion formula? You should cite him or her.

6. In the sampling procedure, line 212 said that the K value was calculated for every seven households... Is it possible to take the K value = 7 for all kebele? To me, the K value is determined by the total population of each kebele. I think the population of each Kebele is varied, as is the K value.

7. There is title redundancy, so it is better to merge them, e.g., lines 221 and 231.

8. There is also a redundancy of sentences, e.g., lines 223 and 244.

9. Line 361... 14.92 What, write clearly.

10. In the discussion parts, the researcher tries to compare with other works in different parts; however, it is without any citation.

Reviewer #3: Please proofread the whole manuscript.

There are grammatical and sentence structure errors throughout the manuscript.

Please check Figure 2. There are names in x-axis that is incomplete.

Suggest to add on in the discussion: how the findings could influence public health policy or healthcare practices within IDP settings.

6. PLOS authors have the option to publish the peer review history of their article (what does this mean? ). If published, this will include your full peer review and any attached files.

**Do you want your identity to be public for this peer review?** For information about this choice, including consent withdrawal, please see our Privacy Policy .

Reviewer #1: No

Reviewer #2: No

Reviewer #3: No

---

## [Author Response · Author response to Decision Letter 1]

5 Dec 2024

Reviewer Comments and Author responses

Reviewer #1:

Abstracts

Comment 1: In Ethiopia now a day there are social unrest, conflict and human made disasters for the sake of this there are displacement of the population from one place to another to relieve this problem and it is the commonest skin diseases of the population lives in internal displaced people. I think it’s well if the authors changes above statements into “In Ethiopia now day there are social unrest, conflict and human made disasters to relieve this problem there are displacement of the population from one place to another”. Also delete the statements “it is the commonest skin diseases of the population lives in internal displaced people and why the authors relate the scabies with conflicts or social unrest, rather its better if the authors relate scabies with hygienic practice, living standards etc. If the research’s conducted in displacement camps, it’s ok.

Thank you, dear reviewer, for your valuable comments and suggestions.

The sentence is revised to make that scabies is more relatable to poor hygiene and sanitation.

Comment 1: Dear researchers I could not see the significance of this study/why this study was needed? Briefly mention your significance. Also merge your study objective with your background.

Dear reviewer,

The document has been corrected based on your comments. The significance of the study has now been briefly mentioned, and the study objective has been merged with the background section as requested

Comment 2: The data was collected by using observation (clinical investigation), structured questionnaires.

Yes, dear reviewer, the data was collected through clinical observation (investigation) and structured questionnaires.

Comment 3: Data was coded, entered and cleaned using with epi data version 3.1, and exported and analysis was done by using SPSS statistical software package version 22. Rearrange above statements, and shorten “Logistic regression analysis was used to identify factors associated with scabies. Findings were presented by using 95% CI of Crude Odds Ratios (COR) and Adjusted Odds Ratios (AOR). To declare statistical significance, a p-value of less than 0.05 was used” Dear Reviewer, thank you for your feedback. The statements have been revised as requested.

Comment 4: in conclusion “Health education on personal hygiene (frequent bathing, washing clothes, and changing clothes regularly), avoiding sharing clothes with others, avoiding contact, sleeping with scabies-ill people, and sharing beds with others”. Simply this list of risk factors, why this factors listed here without any messages. I recommended that the authors mentioned why He list them i.e. his recommendation or the practice of community. Keywords must written in alphabet orders.

Thank you, dear reviewer, for your insightful comment. The list of factors has been clarified to emphasize that they are recommendations aimed at improving community practices related to personal hygiene and reducing scabies transmission. The conclusion now includes a clear statement on the importance of these practices for prevention.

INTRODUCTION

Dear authors it’s interesting if you add the prevalence of scabies in Ethiopia and the study area and also add information about taxonomy of mites and also mention the group organism in which scabies belongs. Dear authors I have seen in your abstracts the scabies is a disease that do not discriminate age, sex, social class etc., but in your introduction there is contrasting idea make clear why this is possible in the introduction.

In this part many sentence is reported without referenced, add necessary references.

Why the authors write some parts of introduction using different text and fonts.

Thank you, dear reviewer, for your valuable comments. The interdiction has been revised accordingly.

Comment 5: Line number 50-51 “Mite is the common cause of this communicable infectious disorder of the skin” change into “Mite is the common causative agents of this communicable infectious disorder of the skin

Comment 6: Line number 51-51: And the mites are not noticeable with the naked eye but can be seen with microscope” delete or add to diagnosis information.

Thank you, dear reviewer, for the suggestion. The sentence has been revised to read.

The sentence on line 51 regarding the mites being not noticeable with the naked eye but visible under a microscope has been deleted, as suggested, or integrated into the diagnosis section for clarity.

Comment 7: Line number 56-57, the onset of the symptoms depends on the level of infection; a scabies patient normally has 12 mites present at any given time. What is the numbers of mites per infected site/wound or the number of mites on/in the body of infected person? Clarify.

Thank you, dear reviewer. The sentence regarding the number of mites in a scabies patient has been deleted to clarify the point, as the specific number of mites per infected site or on/in the body of an infected person was not provided in the context.

Comment 8: Include Line number 99-111, under the first paragraph and also there is repetition of sentence in your introduction, For example line number 82-85 with line number 112-`13.

Thank you for your feedback. The necessary revisions have been made as per your suggestions.

Material and Methods

Dear authors what is the difference b/n the target population and study population, I think it’s good if the authors merge them.

Thank you for the suggestion. The distinction between the target population and the study population has now been clarified, and the two sections have been merged as recommended.

Comment 9: Line number 157: a large majority of displaced persons change the word persons into people and also put the study period at the starts of this parts. Line 171 change the word conducted into employed.

It’s corrected based on your comments. Thank you for the input.

Result

A total of 413 participants were included in the study, making a response rate of 100%. Is your sample size 413 or 422? Make sure. I have seen throughout your result parts the percentage of the response if written before the frequency, also in some area you have written only percentage where as in others both frequency and percentages were reported. Dear authors I recommended you its better if you rewrite the results of your study in similar manners (example 336 (79.6%) or (79.6%), but not possible to write 79.6% (336), although this may not changes what intended to be mentioned, it may reduce the attractiveness and clarity of your paper.

Comment 10: Line 303, more than three-quarters of the sample (79.6%), had a family size greater than or equal to five, rather to say sample, changes to participants, and correct “the majority of 396 (93.8%) of the respondents use wells or springs as a source of water for personal hygiene in the study area” as “the majority of respondents (396 (93.8%) use wells or springs as a source of water for personal hygiene in the study area. Line number 306-307: “Regarding the water source, nearly almost all (93.8%) of households use improved water sources”. Clarify the improved water (i.e. in quality or improved in quantity).

Thank you for your guidance in enhancing the clarity and consistency of the paper. The corrections have been made based on your comments.

Comment 11: Line number 307-308: 7.2% (31) of the study participants have a history of contact with previous scabies patients in the household. Use only percentages, if you include frequency include in all results, but before the percentages outside of the brackets.

DISCUSSION

I appreciated your discussion and for your time to prepare such well organized and informative discussion. But to make your discussion more interesting I have put some recommendation and correction below.

Thank you for your feedback.

The results have been revised for consistency, as recommended.

Corrected accordingly. Thank you for your input.

Comment 12: Line number 316: 14.92 is frequency or percentages/mean and dear authors why you do not use similar significant figures (example 12.93% other hands use 16.4%).

It is done. Thank you for your feedback.

Comment 13: The authors mentioned that your study is comparable with study conducted in Cameroon 17.8% (18), Nigeria 13.3% (19), Arbaminch district, south part of Ethiopia 16.4 (12), Raya Alamata, which is 12.93% (5). Based on what, like your results you have to mention what they reported, here you are list only percentages. Correct as “comparable with study conducted in Cameroon 17.8% (18), Nigeria 13.3% (19), Arbaminch district, south part of Ethiopia 16.4% (12), Raya Alamata, Ethiopia which is 12.93% prevalence (5).

Revised as you suggested. Thank you for the guidance.

Conclusion and Recommendation

Prepared well. Generally the document needed some grammar and language improvement

Thank you for considering these suggestions to enhance the quality and clarity of your research article. I look forward to seeing the revised manuscript.

Grammar and language improvement was made as per you suggestions.

Reviewer #2:

In many place there is typographical errors including the name of Place as example a place that found in the northen parts of Ethiopia called Woldia is not write in its correct name.

Dear Reviewer,

Thank you for your concern. To clarify, it was indeed "Wadila district" from the start, not Woldia.

1. Some ideas are written without any citation (reference), e.g., line 100; line 125 said according to many studies... However, there is no a mentioned reference... So please insert some selective references and another place, so please revise again all introduction parts.

Citation is added

2. Line 149 rearranges the title, like Materials and Methods. Some re-arrangement was done in the Materials and Methods.

3. In the title “Study Setting and Period,” you mentioned the study area, so it requires some modification; moreover, you didn’t explain where the Somali region is found in Ethiopia, e.g., distance from the capital of Ethiopia, in which direction, etc.

Modification was done in the Study Setting.

4. To me, the titles on lines 173 and 175 are the same, so it is better if you merge them. Generally, see again from lines 173-184. Thanks.

Merge was done

5. Who is the honor of the single population proportion formula? You should cite him or her.

Citation is not required because we took 50% estimate, since there is no previous study to cite in the country especially IDP context.

6. In the sampling procedure, line 212 said that the K value was calculated for every seven households... Is it possible to take the K value = 7 for all kebele? To me, the K value is determined by the total population of each kebele. I think the population of each Kebele is varied, as is the K value.

You are correct, we agree you comment and corrected accordingly.

7. There is title redundancy, so it is better to merge them, e.g., lines 221 and 231.

Redundant sentences have been eliminated

8. There is also a redundancy of sentences, e.g., lines 223 and 244.

Redundancy of sentences was removed

9. Line 361... 14.92 What, write clearly.

Correction is done

10. In the discussion parts, the researcher tries to compare with other works in different parts; however, it is without any citation. Citation was included

Reviewer #3:

Please proofread the whole manuscript.

There are grammatical and sentence structure errors throughout the manuscript.

Please check Figure 2. There are names in x-axis that is incomplete.

Suggest to add on in the discussion: how the findings could influence public health policy or healthcare practices within IDP settings.

Thank you for your valuable feedback. We have carefully proofread the manuscript and corrected the grammatical and sentence structure errors. Additionally, we have included a discussion on how the findings could influence public health policy and healthcare practices within IDP settings, as suggested.

---

## [Decision Letter · Decision Letter 1]

29 Dec 2024

PONE-D-24-34114R1PREVALENCE OF SCABIES AND ASSOCIATED FACTORS AMONG SCHOOL AGE CHILDREN IN QOLOJI IDP IN BABILE DISTRICT, SOMALI, EASTERN ETHIOPIA.PLOS ONE

Dear Dr. Ibrahim,

Thank you for submitting your manuscript to PLOS ONE. After careful consideration, we feel that it has merit but does not fully meet PLOS ONE’s publication criteria as it currently stands. Therefore, we invite you to submit a revised version of the manuscript that addresses the points raised during the review process.

Please submit your revised manuscript by Feb 12 2025 11:59PM. If you will need more time than this to complete your revisions, please reply to this message or contact the journal office at plosone@plos.org . Please include the following items when submitting your revised manuscript:

We look forward to receiving your revised manuscript.

Kind regards,

Awatif Abid Al-Judaibi, PhD

Academic Editor

PLOS ONE

Journal Requirements:

Reviewers' comments:

Reviewer's Responses to Questions

**Comments to the Author**

1. If the authors have adequately addressed your comments raised in a previous round of review and you feel that this manuscript is now acceptable for publication, you may indicate that here to bypass the “Comments to the Author” section, enter your conflict of interest statement in the “Confidential to Editor” section, and submit your "Accept" recommendation.

Reviewer #2: All comments have been addressed

Reviewer #3: All comments have been addressed

2. Is the manuscript technically sound, and do the data support the conclusions?

Reviewer #2: Yes

Reviewer #3: (No Response)

3. Has the statistical analysis been performed appropriately and rigorously? 

Reviewer #2: Yes

Reviewer #3: (No Response)

4. Have the authors made all data underlying the findings in their manuscript fully available?

Reviewer #2: Yes

Reviewer #3: (No Response)

5. Is the manuscript presented in an intelligible fashion and written in standard English?

Reviewer #2: Yes

Reviewer #3: (No Response)

6. Review Comments to the Author

Reviewer #2: You try addressed previousely comment, however there is still citation problem. e.g line 301, title 4.4.1 formula etc.

Reviewer #3: (No Response)

7. PLOS authors have the option to publish the peer review history of their article (what does this mean? ). If published, this will include your full peer review and any attached files.

**Do you want your identity to be public for this peer review?** For information about this choice, including consent withdrawal, please see our Privacy Policy .

Reviewer #2: No

Reviewer #3: No

---

## [Author Response · Author response to Decision Letter 2]

14 Jan 2025

Reviewer Comments and Author Responses

Dear Editor and Reviewer,

Thank you for your comments. We have made all the necessary corrections required.

Reviewer #2:

Reviewer Comments: You tried addressing the previous comment; however, there is still a citation problem, e.g., line 301, title 4.4.1 formula, etc.

Author Response: We have corrected this based on your comment. The citations are now grammatically correct.

---

## [Decision Letter · Decision Letter 2]

23 Feb 2025

PREVALENCE OF SCABIES AND ASSOCIATED FACTORS AMONG SCHOOL AGE CHILDREN IN QOLOJI IDP IN BABILE DISTRICT, SOMALI, EASTERN ETHIOPIA.

PONE-D-24-34114R2

Dear Dr. Ahmed Mohammed Ibrahim,

We’re pleased to inform you that your manuscript has been judged scientifically suitable for publication and will be formally accepted for publication once it meets all outstanding technical requirements.

Kind regards,

Awatif Abid Al-Judaibi, PhD

Academic Editor

PLOS ONE

Reviewers' comments:

Reviewer's Responses to Questions

**Comments to the Author**

1. If the authors have adequately addressed your comments raised in a previous round of review and you feel that this manuscript is now acceptable for publication, you may indicate that here to bypass the “Comments to the Author” section, enter your conflict of interest statement in the “Confidential to Editor” section, and submit your "Accept" recommendation.

Reviewer #2: All comments have been addressed

2. Is the manuscript technically sound, and do the data support the conclusions?

Reviewer #2: Yes

3. Has the statistical analysis been performed appropriately and rigorously? 

Reviewer #2: Yes

4. Have the authors made all data underlying the findings in their manuscript fully available?

Reviewer #2: Yes

5. Is the manuscript presented in an intelligible fashion and written in standard English?

Reviewer #2: Yes

6. Review Comments to the Author

Reviewer #2: Currently the peper is in good status. Morover the authors was good to improve the manscruipt. however if you can add the map of the study district

7. PLOS authors have the option to publish the peer review history of their article (what does this mean? ). If published, this will include your full peer review and any attached files.

**Do you want your identity to be public for this peer review?** For information about this choice, including consent withdrawal, please see our Privacy Policy .

Reviewer #2: No

---

## [Editor Report · Acceptance letter]

PONE-D-24-34114R2

PLOS ONE

Dear Dr. Ibrahim,

I'm pleased to inform you that your manuscript has been deemed suitable for publication in PLOS ONE. Congratulations! Your manuscript is now being handed over to our production team.

Kind regards,

on behalf of

Professor Awatif Abid Al-Judaibi

Academic Editor

PLOS ONE